

# Forecasting the cost of drought events in France by super learning

Geoffrey Ecoto[1,2,*] and Antoine Chambaz[2,*]

[1]Caisse Centrale de Réassurance
[2]Université Paris Cité, MAP5 (UMR CNRS 8145)
[*]These authors contributed equally to this work.

**Correspondence:** Geoffrey Ecoto (gecoto@ccr.fr)

**Abstract.** Drought events are the second most expensive type of natural disaster within the French legal framework called the natural disasters compensation scheme. In recent years, droughts have been remarkable in their geographical scale and intensity. We develop a new methodology to forecast the cost of a drought event in France. The methodology hinges on super learning and takes into account the complex dependence structure induced in the data by the spatial and temporal nature of
drought events.

## 1    Introduction

The French state has been facing severe drought events over the past years. The average annual cost of drought events between 2016 and 2020 is 1.1 billion EUROS, a fivefold increase relative to the 2002-2015 period. The recent cycle of extremely intense drought events raises two questions: will climate change perpetuate this pattern (Bradford, 2000; Iglesias et al., 2019) and, if
so, what cost will the French state incur?

In 2015 and 2018 (CCR, 2015, 2018), Caisse Centrale de Réassurance (CCR) launched a study to assess the impact of climate change on the damages caused by natural disasters based on the Intergovernmental Panel on Climate Change (IPCC) scenarios RCP 4.5 and RCP 8.5. Resorting to ARPEGE simulations of the climate in 2050 provided by Météo-France, CCR simulated damages in France in 2050 and concluded that the annual cost in 2050 could increase, depending on the scenario, by
3% (under scenario RCP 4.5) or 23% (under scenario RCP 8.5). Unfortunately, the latter is more likely today than the former.

In (Charpentier et al., 2021), the authors address the problem of predicting the cost of a drought event using Generalized Linear Models (GLM) and tree-based machine learning algorithms. For a given drought event, for each city, the number of claims and the average cost are predicted, then a city-specific predicted cost is obtained by multiplying these two numbers. The aforementioned city-specific predictions exploit several drought indices, topsoil clay concentration, the year, number of
policies and their related insured values, and a binary variable indicating whether the city has already formulated a request for the government declaration of natural disaster.

In our study, like Charpentier et al. (2021), we exploit the Soil Wetness Index (SWI) as a drought indice (it is referred to as the Standardised Soil Water Index by Charpentier et al., 2021). Moreover, we also use sequential cross-validation to take into account the time dependence structure in our data set. In contrast to Charpentier et al. (2021), we rely on a richer description of
the cities obtained by data enrichment (more details to follow). Unlike them, we predict the city-specific costs only for those



cities that have obtained the government declaration of natural disaster (more details on the French legal framework called the natural disasters compensation scheme can be found in (Charpentier et al., 2021, section 2.1)). We emphasize that the problem we tackle is therefore less challenging than theirs. Finally, we make predictions based on a stacking algorithm that adapts the Super Learning methodology to our framework (van der Laan et al., 2007; Benkeser et al., 2018). We call our algorithm the

one-step ahead sequential Super Learner (OSASSL). Its theoretical analysis reveals that the algorithm can efficiently learn from our data set, a *short* time-series whose time-specific observations consist of a large network of slightly dependent data (Ecoto et al., 2021).

**Organization of the article.**

In Section 2, we present the data that we collected and used in this study. In Section 3, we describe the OSASSL. In Section 4,

we expose and comment on the results that we obtain. In Section 5, we discuss directions for future work.

## 2   Data

We merge several data sets into a master data set. The merged data sets are either provided by CCR's cedents or collected by us from other sources. They contribute different kinds of information.

Of note, in the rest of this study, France refers to *Metropolitan* or *Mainland* France. Drought events are not a threat in

Overseas France (essentially because there is little clay in these parts of the country).

### 2.1   Data provided by CCR's cedents

Ninety-five percents of the French natural disasters insurance market is reinsured by CCR. Contractually, its cedents must share their portfolios and claims data. Over the years, CCR has thus gathered a vast collection of locations and characteristics of insured goods and claims data. From 1990 to present, the collection covers roughly 22% to 97% of all the French claims.

### 2.2   Data garnered from other sources

The data set, based so far on data provided by cedents only, is then enriched with data from other sources, namely the National Institute for Statistical and Economic Studies (INSEE), the Geographic National Institute (IGN), the French Geological Survey (BRGM) and Météo-France. The new features supplementing the description of the cities are seismic hazard and climatic zone, clay shrinkage-swelling hazards, tree coverage rate, area, population and years of construction. Lastly, we benefit from the Soil

Wetness Index (SWI) as described in (Charpentier et al., 2021, section 2.3).

### 2.3   City-level data processing

Some data are available at the house-level (namely, the cost of claim and insured sum), but most are not. In particular, the SWI is available at a $8 \times 8$ km$^2$ resolution, while the 90%-quantile of the French cities area is 30 km$^2$. Consequently, we choose to work at a city level and thus aggregate the features that have a higher resolution. Details follow.





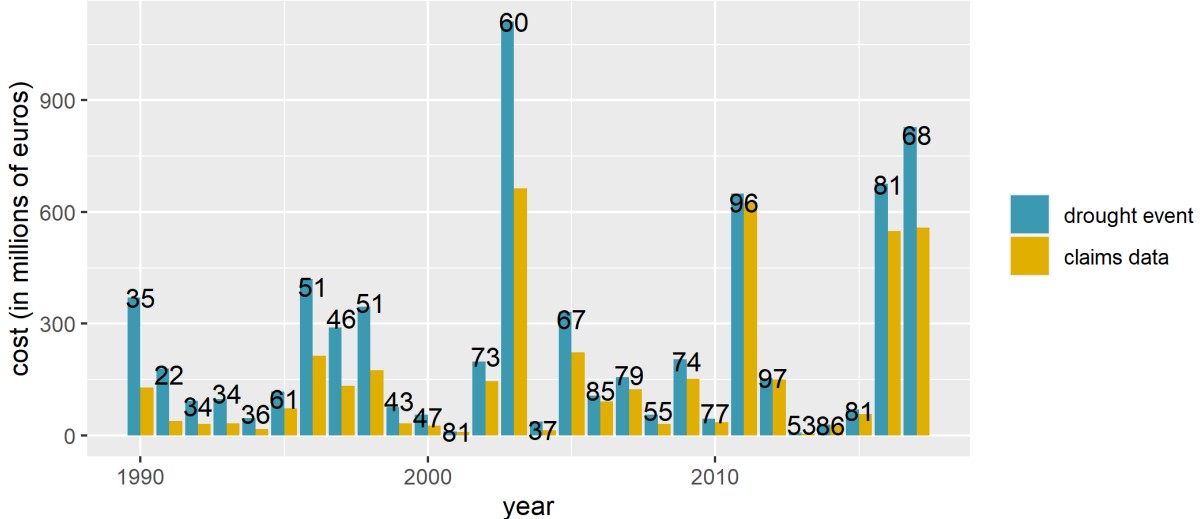

**Figure 1.** Estimated overall costs of drought events across France (blue) and provisional city-specific costs obtained by aggregating the costs of those claims filled in the claims data provided by the cedents (yellow). The ratios of the latter to the former range between 22% and 97%.

**About the city-level costs of drought events.**

The cost of the damages in a city caused by a drought event (what will be our response variable) is unknown. However, on the one hand the overall cost across France is estimated by actuarial studies and, on the other hand, we know the costs *of those* claims filled in the claims data provided by the cedents which, unfortunately, only represent a fraction of all the claims.

Provisional city-specific costs are computed by aggregating by city the costs filled in the claims data provided by the cedents. Because these claims data are not exhaustive, the sum of all the provisional city-specific costs is smaller than the estimated overall cost. The (final) city-specific costs are proportional to the provisional city-specific costs in such a way that the sum of all the (final) city-specific costs equals the estimated overall cost.

Figure 1 illustrates the gaps between the estimated overall costs across France and the sum of the provisional city-specific costs. The ratios of the latter to the former range from 20% to 90%.

**About the city-level SWI.**

For every year and every city, we derive a collection of 36 city-level SWIs, one for each ten-day period (a *décade* in French) that make up a year. Each of these 36 SWIs is the convex average of the corresponding SWIs of the $8 \times 8$ km$^2$ squares that overlap the city's area. The weights are proportional to the areas of the intersections.

We use the city-level SWIs to build a rich collection of SWI-related covariates.





Because the effects of a drought event can build up slowly, for every year $t$ and every city, we concatenate the $3 \times 36$ ten-day city-level SWIs of years $t$, $(t-1)$ and $(t-2)$. We also add the minima, means and standard deviations of the 36 ten-day city-level SWIs computed separately over the years $t$, $(t-1)$ and $(t-2)$.

In addition, for every year $t$ and every city, we compute and concatenate the mean SWI of all ten-day periods from April to September for *(a)* year $t$ alone, *(b)* years $t$ and $(t-1)$, *(c)* years $t, (t-1)$ and $(t-2)$.

Moreover, for each quarter $1 \leq q \leq 4$ (January-March, April-June, July-September, October-December), for every year $\tau$ between 1959 and 2017 and every city $\alpha$, we compute the average city-level SWI, $\overline{\text{SWI}}_{q,\tau,\alpha}$ and form the four cumulative distribution functions $\hat{F}_q$ associated to the four data sets $\{\overline{\text{SWI}}_{q,\tau,\alpha} : 1959 \leq \tau \leq 2009, \alpha\}$. Then, for every year $1990 \leq t \leq 2017$ and every city $\alpha$, we also add the $3 \times 4$ probabilities $\hat{F}_q(\overline{\text{SWI}}_{q,t,\alpha}), \hat{F}_q(\overline{\text{SWI}}_{q,t-1,\alpha}), \hat{F}_q(\overline{\text{SWI}}_{q,t-2,\alpha})$ $(q = 1, \ldots, 4)$.

**About the city-level description.**

For every year, each city is described by a collection of covariates. A city's multi-faceted description attempts to capture all the city's traits that, beyond the city-level SWIs presented in the previous paragraph, can explain the cost of a possible drought event. It contains:

–   the year $t$;

–   the city's area, average altitude, climatic zone (a five-category variable), seismic zone (a four-category variable), propor-
tions of surface with a "tree-coverage" greater than 10%, where "tree-coverage" corresponds to a wooded area, a wood, a vineyard, a hedge, a poplar grove, a woody heath, an orchard, a coniferous forest, an open forest, a closed forest of mixed trees, a closed forest of deciduous trees, and any of the 11 above types of terrain;

–   the number of inhabitants, (estimated) number of houses located within the city's limits, house density, defined as the ratio of the number of houses to the city's area, proportions of buildings built prior to 1949, between 1950 and 1974,
between 1975 and 1989, and after 1989;

–   the proportions of the houses located within the city's limits that fall in each of the four clay shrinkage-swelling hazards categories;

–   the (estimated) insured sum corresponding to the houses located within the city's limits, the average house value, defined as the ratio of the aforementioned insured sum to the number of houses;

–   five indicators of whether or not the municipality formulated a request for the government declaration of natural disaster on years $t$ to $(t-4)$, and five indicators of whether or not the municipality obtained the government declaration of natural disaster on years $t$ to $(t-4)$.

In addition, the city's description contains:

–   the cumulated city-level costs computed across the years $(t-1)$ to $(t-5)$;





– the mean and median city-level costs of the drought events computed across the years $(t-1)$ to $(t-5)$ and all cities within the same department (one of the three levels of government under the national level, between the administrative regions and the communes; metropolitan France counts 96 departments).

The city's description is finally enriched with *compound covariates*. The compound covariates have a similar form. For every year $t$ and each city $\alpha$, for a given covariate $C_{h,\alpha,t}$ defined for all houses $h$ within the city's limits (and in the portfolios data

provided by the cedents), we compute the weighted mean $\sum_h s_{h,\alpha,t} \times C_{h,\alpha,t} / \sum_h s_{h,\alpha,t}$ where $s_{h,\alpha,t}$ is the (estimated) insured sum of house $h$ located within $\alpha$'s limits on year $t$. Here $C_{h,\alpha,t}$ can be:

1. the mean of all the $t$-specific 36 ten-day SWIs of the $8 \times 8$km$^2$ square which contains house $h$;

2. the level of the clay shrinkage-swelling hazard localized at $h$ (does not depend on $t$);

3. the ground slope localized at $h$ (does not depend on $t$);

4. the three products of (1) and (2), (1) and (3), (1), (2) and (3).

Moreover, for every year $t$ and every city $\alpha$, for each $C_{h,\alpha,t}$ among (1), (2) and (3), we also add the 30 29-quantiles of the data set $\{s_{h,\alpha,t} \times C_{h,\alpha,t} : h\}$ (where $h$ ranges over the set of houses $h$ within $\alpha$'s limits).

## 3 The one-step ahead sequential Super Learner (OSASSL)

### 3.1 Presentation and theoretical performance

In (Ecoto et al., 2021), we developed and studied the so-called OSASSL, an algorithm designed to learn a (stationary) feature of the law of a *short* time-series with time-specific observations consisting of *many* dependent data-structures. The algorithm builds upon the canonical Super Learning methodology (van der Laan et al., 2007). Its analysis reveals that our algorithm manages to make up for the *shortness* of the time-series thanks to the *manyness* of each time-specific observation provided that the latter are only slightly dependent. Here, we present an instance of the OSASSL specifically built to forecast the cost of

drought events.

We let $(\bar{O}_t)_{t \geq 1}$ denote the time-series that formalizes the time-series described in section 2. At each time $t \in \mathbb{N}^*$, $\bar{O}_t$ consists of a finite collection $(O_{\alpha,t})_{\alpha \in \mathcal{A}}$ of $(\alpha,t)$-specific observations, where each $\alpha \in \mathcal{A}$ represents a French city. For every $\alpha \in \mathcal{A}$, $O_{\alpha,t}$ decomposes as $O_{\alpha,t} := (Z_{\alpha,t}, X_{\alpha,t}, Y_{\alpha,t}) \in \mathcal{Z} \times \mathcal{X} \times [0, B] =: \mathcal{O}$ where $X_{\alpha,t} \in \mathcal{X}$ is the collection of covariates describing the city $\alpha$ on year $t$, $Z_{\alpha,t} \in \mathcal{Z}$ is the city-level SWI describing the drought event that year, and $Y_{\alpha,t} \in [0, B]$ is the city-specific

cost of the drought event that year (known to take its values between 0 and a constant $B$). We assume that the mean conditional cost $\theta^\star : (x, z) \mapsto \mathbb{E}[Y_{\alpha,t}|X_{\alpha,t} = x, Z_{\alpha,t} = z]$ does not depend on $(\alpha, t)$ or, in other terms, that it is a stationary feature of the law of $(\bar{O}_t)_{t \geq 1}$. This is the case if, given a time-specific city-description and SWI $(X_{\alpha,t}, Z_{\alpha,t})$, the mechanism that produces a cost after a drought event conditionally on $(X_{\alpha,t}, Z_{\alpha,t})$ does not depend on $(\alpha, t)$, that is, remains constant throughout time and France.





In this project the OSASSL is a meta-algorithm that learns the mean conditional cost $\theta^\star$ from $(\bar{O}_t)_{t\geq 1}$ by aggregating the estimators of $\theta^\star$ provided by a user-supplied collection of $J$ algorithms $\widehat{\theta}_1, \ldots, \widehat{\theta}_J$. At each time $t \geq 1$, every algorithm $\widehat{\theta}_j$ trained on $\bar{O}_1, \ldots, \bar{O}_t$ outputs an estimator $\theta_{j,t}$ of $\theta^\star$. The OSASSL selects the best algorithm indexed by $\widehat{j}_t$ defined as the minimizer of the empirical average cumulative risks,

$$\widehat{j}_t \in \underset{1 \leq j \leq J}{\arg\min} \, \widehat{R}_{j,t}, \tag{1}$$

where

$$\widehat{R}_{j,t} := \frac{1}{t|\mathcal{A}|} \sum_{\tau=1}^{t} \sum_{\alpha \in \mathcal{A}} \left[ Y_{\alpha,\tau} - \theta_{j,\tau-1}(X_{\alpha,\tau}, Z_{\alpha,\tau}) \right]^2. \tag{2}$$

Interestingly, the OSASSL is an online algorithm if each of the $J$ algorithms $\widehat{\theta}_1, \ldots, \widehat{\theta}_J$ is online, that is, such that the making of $\theta_{j,t}$ consists in an update of $\theta_{j,t-1}$ based on newly accrued data $\bar{O}_t$.

The $t$-specific measure of performance of each $\widehat{\theta}_j$ is the unknown quantity

$$\widetilde{R}_{j,t} := \frac{1}{t|\mathcal{A}|} \sum_{\tau=1}^{t} \sum_{\alpha \in \mathcal{A}} \mathbb{E} \left\{ \left[ Y_{\alpha,\tau} - \theta_{j,\tau-1}(X_{\alpha,\tau}, Z_{\alpha,\tau}) \right]^2 \Big| \bar{Z}_\tau, F_{\tau-1} \right\} \tag{3}$$

where $F_t$ is the history generated by $\bar{O}_1, \ldots, \bar{O}_t$ (by convention, $F_0 = \emptyset$). It takes the form of an average cumulative risk conditioned on the sequence $(\bar{Z}_t)_{t\geq 1}$ with $\bar{Z}_t = (Z_{\alpha,t})_{\alpha \in \mathcal{A}}$. The $t$-specific oracular meta-algorithm is indexed by the oracular $\widetilde{j}_t$ defined as the minimizer

$$\widetilde{j}_t \in \underset{1 \leq j \leq J}{\arg\min} \, \widetilde{R}_{j,t}, \tag{4}$$

which, like each $\widetilde{R}_{j,t}$, is unknown to us. Note that $\widehat{R}_{j,t}$ estimates $\widetilde{R}_{j,t}$ and that (1) mimics (4).

The theoretical analysis hinges on a key-assumption about the dependence structure in the time-series $(\bar{O}_t)_{t\geq 1}$. We exploit conditional dependency graphs to model the amount of conditional independence. Specifically, we assume the existence of a graph $\mathcal{G}$ with vertex and edge sets $\mathcal{A}$ and $\mathcal{E}$ such that if $\alpha \in \mathcal{A}$ is not connected by any edge $e \in \mathcal{E}$ to any $\alpha' \in \mathcal{A}' \subset \mathcal{A}$, then $O_{\alpha,t}$ is conditionally independent of $(O_{\alpha',t})_{\alpha' \in \mathcal{A}'}$ given $F_{t-1}$ and $\bar{Z}_t$. Then what matters is how much connected is the graph, as reflected by its degree, $\deg(\mathcal{G})$, which equals 1 plus the largest number of edges that are incident to a vertex in $\mathcal{G}$. Finally, let us emphasize that the dependency graph $\mathcal{G}$ plays no role in the OSASSL's characterization and training. However, it is pivotal in the algorithm's theoretical analysis.

The performance of $\widehat{j}_t$ as an estimator of $\widetilde{j}_t$ is expressed in terms of a comparison of the excess risk of the former to the excess risk of the latter. Under additional mild assumptions (Ecoto et al., 2021, corollary 2) there exists a decreasing function $C : \mathbb{R}_+^* \to \mathbb{R}_+^*$ such that, for any $\varepsilon > 0$,

$$\mathbb{E} \left[ \underbrace{\widetilde{R}_{\widehat{j}_t,t} - \widetilde{R}_t(\theta^\star)}_{\text{excess risk of } \widehat{j}_t} - (1+\varepsilon) \Big( \underbrace{\widetilde{R}_{\widetilde{j}_t,t} - \widetilde{R}_t(\theta^\star)}_{\text{excess risk of } \widetilde{j}_t} \Big) \right] \leq C(\varepsilon) \frac{\log(J \log(\mathcal{I}^2))}{\mathcal{I}^2} \tag{5}$$





| min. | 1st qu. | median | mean | 3rd qu. | 99%-qu. | max |
|------|---------|--------|------|---------|---------|-----|
| 0 | 5 | 6 | 5.96 | 7 | 11 | 29 |

**Table 1.** Quartiles, 99%-quantile and mean of the numbers of neighboring cities in France in 2019. Although the maximum cannot be interpreted literally as $\deg(\mathcal{G}) - 1$, it nevertheless gives a sense of what a meaningful value of $\deg(\mathcal{G})$ can be.

where $\mathcal{I}^2$ grows like the amount of information available and can be equal to either $t$ or $|\mathcal{A}|/(t \deg(\mathcal{G}))$. If the ratio $|\mathcal{A}|/\deg(\mathcal{G})$ is sufficiently large (both in absolute terms and relative to $t$), then the oracular inequality (5) is sharper when $\mathcal{I}^2 = |\mathcal{A}|/(t \deg(\mathcal{G}))$ than when $\mathcal{I}^2 = t$. This reveals that the OSASSL can leverage a large ratio $|\mathcal{A}|/\deg(\mathcal{G})$ in the face of a small $t$.

In the application, $t \approx 25$, $|\mathcal{A}| \approx 36,000$. As for $\deg(\mathcal{G})$, it is much harder to assess a meaningful value. In this regard, it is relevant to recall that, in 2019, France had around $1,000$ federations of municipalities, each regrouping 30 cities on average. Furthermore, we computed the number of neighboring cities for each city. The quantiles and mean of these numbers are reported in Table 1. In particular, the city with the largest number of neighboring cities (Paris) has 29 of them.

### 3.2 Forecasting the cost of drought events

The OSASSL presented in Section 3.1 is designed to learn the mean conditional cost $\theta^\star$ from $(\bar{O}_t)_{t \geq 1}$. At each time $t \geq 1$, it outputs the $t$-specific estimator $\theta_{\hat{j}_t, t}$. This estimator can be evaluated at every $(X_{\alpha, t+1}, Z_{\alpha, t+1})$ $(\alpha \in \mathcal{A})$ and we use the sum

$$\sum_{\alpha \in \mathcal{A}} \theta_{\hat{j}_t, t}(X_{\alpha, t+1}, Z_{\alpha, t+1})$$

to predict the cost of the drought event at time $(t+1)$, that is, $\sum_{\alpha \in \mathcal{A}} Y_{\alpha, t+1}$.

## 4 Application

This section discusses the practical implementation, training and exploitation of the OSASSL presented and studied in Section 3. Section 4.1 describes the collection of $J$ algorithms $\hat{\theta}_1, \ldots, \hat{\theta}_J$; Section 4.2 explains how the OSASSL is trained; Section 4.3 presents the results and comments upon them.

### 4.1 Implementing two OSASSLs

In fact, we deploy two meta-algorithms taking the form of OSASSLs, the so-called discrete and continuous overarching Super Learners. Both rely on the same library of $J$ algorithms $\hat{\theta}_1, \ldots, \hat{\theta}_J$. These $J$ algorithms are themselves OSASSLs either in the strict or in a loose sense – more details to follow.



**Penalization.**

Because our ultimate goal is to forecast the cost of the latest drought event, we made the decision to rely on a penalized version of $\widehat{R}_{j,t}$ (2), by substituting

$$\widehat{R}_{j,t} + \frac{0.05}{t} \sum_{\tau=1}^{t} \Big( \underbrace{\sum_{\alpha \in \mathcal{A}} Y_{\alpha,\tau}}_{\text{actual cost}} - \underbrace{\sum_{\alpha \in \mathcal{A}} \theta_{\widehat{j}_{\tau-1},\tau-1}(X_{\alpha,\tau}, Z_{\alpha,\tau})}_{\text{predicted cost}} \Big)^2 \tag{6}$$

for $\widehat{R}_{j,t}$ (we recall that $\theta_{j,t}$ is the output of $\widehat{\theta}_j$ trained on $\bar{O}_1, \ldots, \bar{O}_t$ and that $\widehat{j}_t$ is defined in (1)). Observe that each $t$-specific penalization term equals 0.05 times the average over $1 \leq \tau \leq t$ of the $\tau$-specific squared difference between the actual cost of the drought event (left-hand side summand) and the predicted cost made by the (penalized) OSASSL trained on $\bar{O}_1, \ldots, \bar{O}_{\tau-1}$ (right-hand side summand). The factor 0.05 was chosen somewhat arbitrarily.

By adding this penalization term, the OSASSL favors the algorithms that better predict not only the *city-specific* costs but also the *overall* cost of the next drought event. In addition, the penalization term slightly dilutes the importance of the city-specific costs and, on the contrary, reinforces the importance of the overall cost, the latter being more dependable than the formers as we explained in Section 2.3.

**The discrete and continuous overarching Super Learners.**

Called the *discrete* overarching Super Learner, the first OSASSL is the algorithm that, at time $t \geq 1$, outputs $\theta_{\widehat{j}_t,t}$ (using (6) instead of (2) as an empirical measure of the risk).

We also consider a second OSASSL which is defined as a regular OSASSL based on a library derived from $\widehat{\theta}_1, \ldots, \widehat{\theta}_J$ and comprising $J' = O(\varepsilon^{1-J})$ algorithms where $\varepsilon > 0$ is a small positive number ($J' = O(\varepsilon^{1-J})$ means that $J'$ is upper-bounded by a constant times $\varepsilon^{1-J}$). Specifically, these $J'$ algorithms are denoted by $\widehat{\boldsymbol{\theta}}_\pi$ where the index $\pi$ ranges in an $\varepsilon$-net over the simplex $\{x \in (\mathbb{R}_+)^J : \sum_{j=1}^J x_j = 1\}$ (an $\varepsilon$-net whose cardinality is $J'$, *i.e.*, a finite subset of $J'$ elements of the simplex which "approximates" the simplex). For each $\pi$ in the $\varepsilon$-net, $\widehat{\boldsymbol{\theta}}_\pi$ trained on $\bar{O}_1, \ldots, \bar{O}_t$ outputs the $\pi$-specific convex combination $\sum_{j=1}^J \pi_j \theta_{j,t}$. The bound in (5) is still meaningful when $\varepsilon = O(\mathcal{I}^{-1})$. We refer to this second OSASSL as the *continuous* overarching Super Learner.

**Their library of algorithms.**

We now turn to the description of the $J$ algorithms $\widehat{\theta}_1, \ldots, \widehat{\theta}_J$. All of them rely on a collection of base learners $\widehat{\mathcal{L}}_1, \ldots, \widehat{\mathcal{L}}_K$. Some of the base learners rely on linear models and their extensions (lasso, ridge, elastic net, multivariate adaptive regression splines, support vector regression). Others are tree-based algorithms (CART, random forest, gradient boosting), or rely on neural networks. Others fall in the category of $k$-nearest-neighbors algorithms tailored to our study so that the dissimilarity between observations is a convex combination of the Kolmogorov-Smirnov distances between the empirical cumulative distribution functions mentioned in Section 2.3. Finally, some are regular Super Learners themselves, based on a selection of the aforementioned base learners and oblivious to the temporal ordering (*i.e.*, they rely on vanilla inner cross-validation).



Moreover, some of these base learners are combined (upstream) with so-called screening algorithms. A screening algorithm is merely an algorithm that selects a subset of the covariates deemed relevant to feed the base learners. In general, the selection can be either deterministic or data-driven. In our study, we only use deterministic screening algorithms based on expert knowledge.

Overall, we implement a collection of $K = 27$ base learners (including the variants obtained by combining with different screening algorithms). The collection is shared by the $J$ algorithms $\widehat{\theta}_1, \ldots, \widehat{\theta}_J$ which differ in the methods they rely on to exploit the base learners.

One of the method yields a OSASSL precisely as defined in (1) and (2)/(6) where we substitute $K$ for $J$ and $\ell_{j,\tau-1}$ for $\theta_{j,\tau-1}$, with $\ell_{j,t}$ the output of $\widehat{\mathcal{L}}_j$ trained on $\bar{O}_1, \ldots, \bar{O}_t$. The resulting OSASSL is an instance of discrete Super Learner as previously described when introducing the first overarching Super Learner. As we already explained, the library of base learners $\widehat{\mathcal{L}}_1, \ldots, \widehat{\mathcal{L}}_K$ can be extended using an $\varepsilon$-net over the simplex $\{x \in (\mathbb{R}_+)^K : \sum_{k=1}^K x_k = 1\}$. For each $\pi$ in the $\varepsilon$-net, $\widehat{\mathcal{L}}_\pi$ trained on $\bar{O}_1, \ldots, \bar{O}_t$ outputs the $\pi$-specific convex combination $\sum_{k=1}^K \pi_k \ell_{k,t}$. Using the extended collection of base learners, the same method then yields an instance of continuous Super Learner as previously described when introducing the second overarching Super Learner.

In a similar fashion, we consider several methods to exploit the base learners $\widehat{\mathcal{L}}_1, \ldots, \widehat{\mathcal{L}}_K$. Heuristically, the principle is to learn to produce a single prediction based on the multiple predictions made by the base learners once they have been trained, just like we described in the previous paragraph. Some methods rely on the same method as above with an extra penalization term in the definition of the risk (similar to the one used to define (6) based on (2)). The other methods rely on the lasso, ridge and elastic net algorithms, or on the random forests, gradient boosting and support vector regression algorithms. Finally, some of the methods can exploit the covariates. Overall, we implement a collection $J = 48$ algorithms $\widehat{\theta}_1, \ldots, \widehat{\theta}_J$.

## 4.2 Training

At each time $t \geq 1$ we define a summary of the past based on observations made during the five previous years. This is very relevant for two reasons. First, a drought-related claim can be the by-product of repeated shrinkage-swelling episodes over the years. Second, a city-level cost of a drought event is expected to be high when the city did not benefit recently from a declaration of natural disaster (because of the possible accumulation of damages over the years); on the contrary, it is expected to be low otherwise (because damages may already have been compensated). To do so, we reserve the data from year 1990 to year 1994.

For each $t \in \{1995, \ldots, 1999\}$, we derive $\ell_{1,t-1994}, \ldots, \ell_{K,t-1994}$. For each $t \in \{2000, \ldots, 2005\}$, we derive $\theta_{1,t-1994}, \ldots, \theta_{J,t-1994}$ using $\ell_{1,(t-1)-1994}, \ldots, \ell_{K,(t-1)-1994}$, and also $\ell_{1,t-1994}, \ldots, \ell_{K,t-1994}$. For each $t \in \{2006, \ldots, 2017\}$, we derive the discrete overarching Super Learner $\widehat{j}_{t-1994}$ using $\theta_{1,(t-1)-1994}, \ldots, \theta_{J,(t-1)-1994}$ (which rely themselves on $\ell_{1,(t-2)-1994}, \ldots, \ell_{K,(t-2)-1994}$), and also $\theta_{1,t-1994}, \ldots, \theta_{J,t-1994}$ and $\ell_{1,t-1994}, \ldots, \ell_{K,t-1994}$. For each $t \in \{2006, \ldots, 2017\}$, the continuous overarching Super Learner is derived too.

We cannot train our algorithms beyond the year 2017 because, to this day, the real costs and city-level costs are still too uncertain.





The numerical analysis was conducted in R (R Core Team, 2022). We adapted the R package `SuperLearner` (Polley et al., 2021) in a package called `SequentialSuperLearner` (Chambaz and Ecoto, 2021).

### 4.3 Results

In Figure 2 we present the evolution of the weights that characterize the continuous overarching Super Learner through the
years 2007 to 2017. The figure reveals that only four of the $J = 48$ algorithms $\widehat{\theta}_1, \ldots \widehat{\theta}_J$ get a positive weight, and that only two of them do in 2016 and 2017. Moreover, one of the algorithms dominates the others during the whole training. It does not come as a surprise that this algorithm (whose method is a variant of gradient boosting with linear boosters) is constantly selected by the discrete overarching Super Learner.

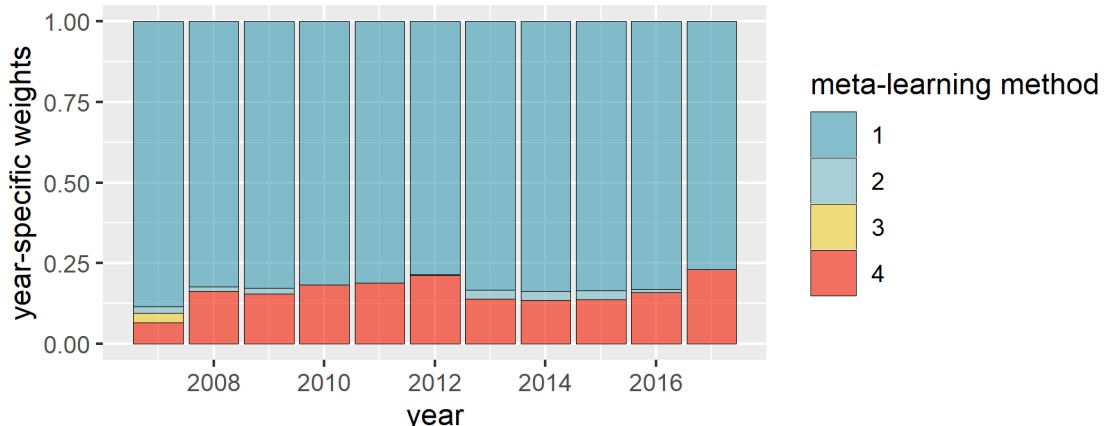

**Figure 2.** Evolution (from 2007 onward) of the weights attributed in the overarching Super Learner to four of the algorithms $\widehat{\theta}_1, \ldots \widehat{\theta}_J$. The others get no weight at all.

For confidentiality reasons, we were not given the authorization to discuss how the overarching Super Learners fare compared
to the algorithm currently deployed at CCR to predict the overall costs of drought events in France from 2007 to 2017. However, we were authorized to make a comparison for the sole year 2017. That particular year, the discrete and continuous overarching Super Learners outperform the algorithm currently deployed at CCR, with a precision of 96% (discrete overarching Super Learner), 94% (continuous overarching Super Learners) versus 83% (currently deployed algorithm).

In Figure 3 we present three sequences of predictions from 2007 to 2017: those from the discrete and continuous overarching
Super Learners and those obtained by averaging all the base learners' predictions (for comparison). Note that the two sequences of predictions from the Super Learners are quite similar. Overall, the Super Learners' predictions are generally accurate and better than the averaged predictions. In Table 2 we report the averages and standard deviations (over the years) of the ratios of the predicted costs to the real costs for the predictors. Both in terms of mean and standard deviation, the discrete overarching Super Learner outperforms its continuous counterpart, which itself outperforms the predictor that averages all the base learners'





predictions. Furthermore, the two Super Learners' predictions are quite good for all years except 2012 and 2016. The poorer predictions in 2016 are more problematic because the real cost in 2016 is much higher than in 2012.

| predictions | mean | standard deviation |
|---|---|---|
| average of the base learners' predictions | 1.21 | 0.42 |
| continuous overarching Super Learner's | 1.10 | 0.32 |
| discrete overarching Super Learner's | 1.04 | 0.28 |

**Table 2.** Averages and standard deviations (over the years) of the ratios of the predicted costs to the real costs. The predictions are either those made by the discrete and continuous overarching Super Learners or obtained by averaging all the base learners' predictions.

The year 2016 is known in the French insurance market as particularly challenging. Unfortunately, as far as we know, this fact is undocumented in the literature. However, we can report two facts to uphold this statement. First, the year-specific average cost (meaning here the ratio of the total cost of the year's drought event to the corresponding number of declarations of natural

disaster delivered that year) is particularly large in 2016 compared to the global average cost (meaning here the ratio of the total cost of the drought events between 2007 and 2017 to the total number of declarations of natural disaster delivered these years): 797,000 EUROS versus 482,000 EUROS. Second, we can quote Charpentier et al. (2021) who say of their predictions for the year 2016 that they are "severely underestimated". Judging by their Figure 7, the underestimation by the discrete and continuous overarching Super Learners for the year 2016 is less pronounced than the underestimation by their algorithms (but

we recall that they tackle a more challenging problem than us because we focus on the city-specific costs for those cities that have obtained the government declaration of natural disaster whereas they consider all French cities).

In Figure 4 we present (Gaussian) kernel density estimates of the conditional laws of the residual error (defined as the real cost minus the prediction made by the continuous overarching Super Learner – the figure is very similar when substituting the discrete overarching Super Learner for the continuous one) in ten strata characterized by the deciles of the city-level costs. We

note that the higher the city-level costs, the higher the residuals. Moreover, the overarching Super Learner tends to overestimate the costs in cities with lower city-level costs and, on the contrary, it tends to underestimate them in cities with higher city-level costs.

In Figure 5 we present two maps that provide insight into the geographical distribution of the residual errors (of the predictions made by the continuous overarching Super Learner – the maps are very similar when considering its discrete counterpart).

In the left-hand side map, a city contributes as many points as the number of times it benefited from a declaration of natural disaster between 2007 and 2017. In the right-hand side map, a city contributes a point if and only if it benefited from a declaration of natural disaster in 2016, the year considered as particularly challenging. In both maps, the color reflects the quartile of the residual error to which the city- and time-specific residual error belongs. Moreover, in the left-hand side map the transparency reflects the number of times the city benefited from a declaration of natural disaster between 2007 and 2017, a larger number

leading to less transparency. By comparing the two maps, we notice *(i)* that the 2016 drought episode impacted very strongly the South of France and *(ii)* that, in this region, the residual errors tend to be higher, leading to the underestimation of the local cost.





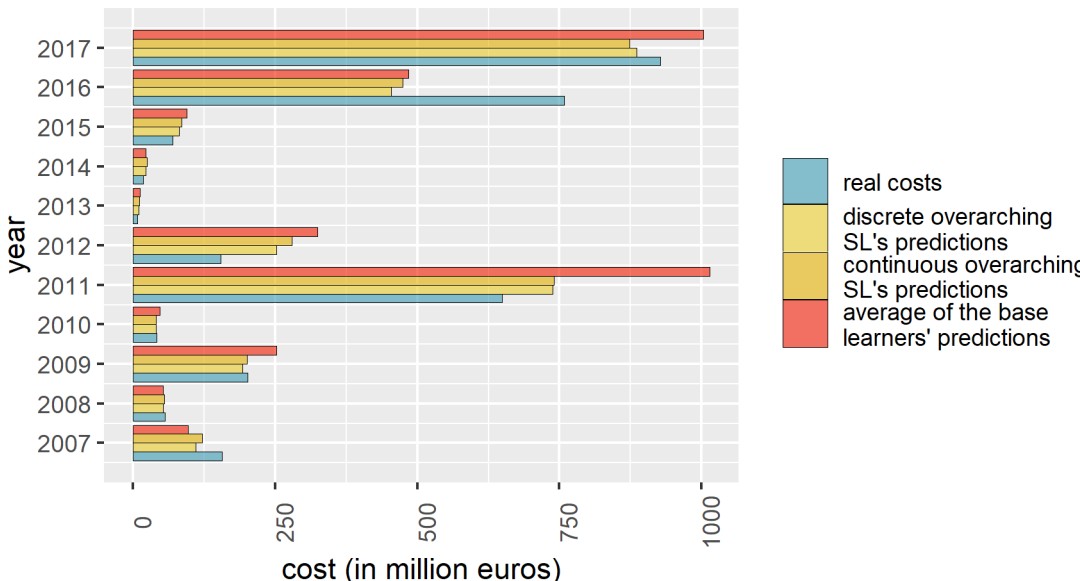

**Figure 3.** Presentation (from 2007 onward) of the real costs of drought events and their predictions. The predictions are either those made by the discrete and continuous overarching Super Learners or obtained by averaging all the base learners' predictions.

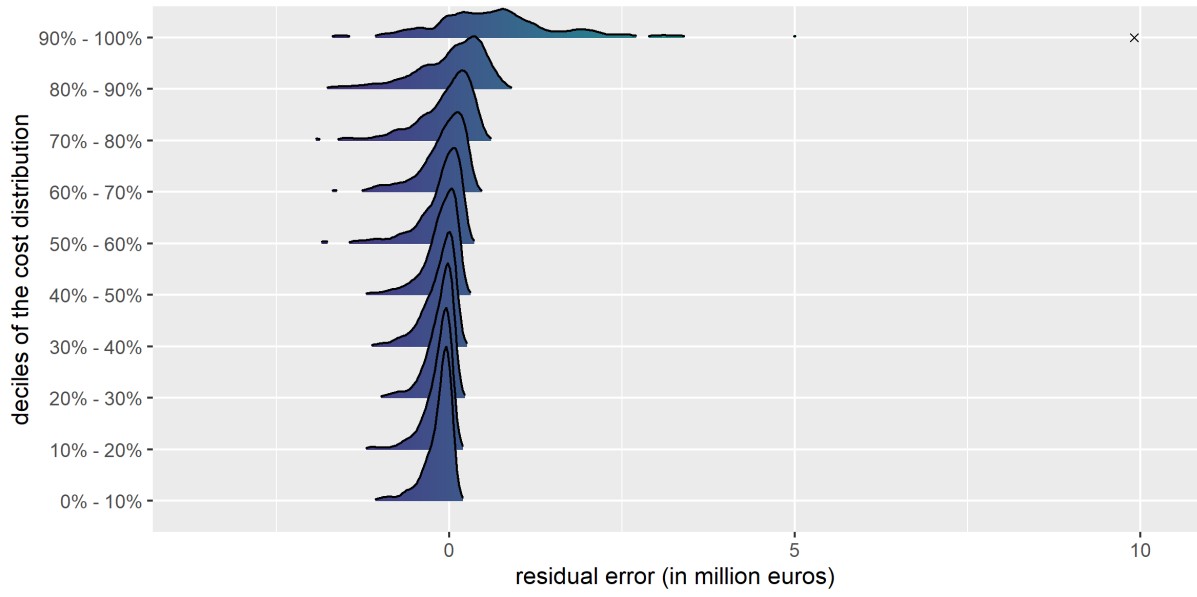

**Figure 4.** Kernel density estimates of the conditional laws of the residual error (of the predictions made by the continuous overarching Super Learner) in ten strata characterized by the deciles of the city-level costs. The cross at the upper right-hand side of the plot indicates the maximum residual error, made for a city belonging to the last decile of the cost distribution.



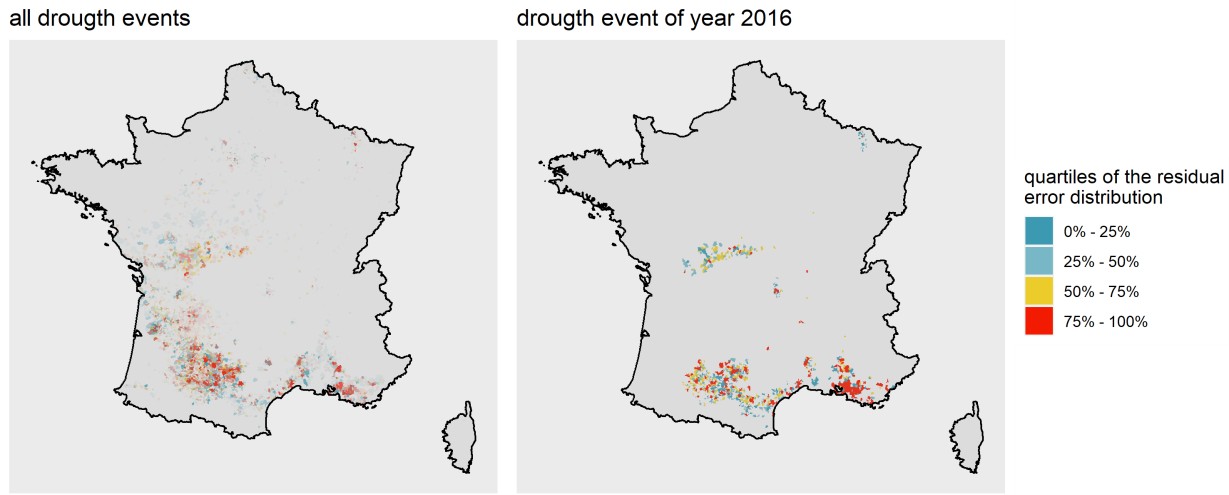

**Figure 5.** Geographical distribution of the residual errors (of the predictions made by the continuous overarching Super Learner). Left-hand side map: a city contributes as many points as the number of times it benefited from a declaration of natural disaster between 2007 and 2017. Right-hand side map: a city contributes a point if and only if it benefited from a declaration of natural disaster in 2016. The color reflects the quartile of the residual error to which the city- and time-specific residual error belongs (based on all the errors). In the left-hand side map, the transparency reflects the number of times the city benefited from a declaration of natural disaster between 2007 and 2017, a larger number leading to less transparency.

## 5    Discussion

The French legal framework known as the natural disasters compensation scheme was created in 1982. Drought events were included in 1989 and have been since then the second most expensive type of natural disaster. In recent years, drought events have been remarkable in their extent and intensity. The problem is worsening and not limited to France, as was predicted in the technical report (Wüest et al., 2011, page 7): "as our climate continues to change, the risk of property damage from soil subsidence [*i.e.*, drought events] is not only increasing but also spreading to new regions across Europe".

Forecasting the cost of a drought event is important actuarial problem. To tackle this challenge, we develop a new methodology that builds upon super learning. Our so-called overarching Super Learner blends predictions made by a collection of OSASSLs which, themselves, blend the predictions made by a variety of machine-learning algorithms.



In (Ecoto et al., 2021) we studied the theoretical properties of the overarching Super Learner. We showed that it can learn despite the complex dependence structure induced in the data by the spatial and temporal nature of the phenomenon of drought, making up for the shortness of the time-series thanks to the manyness of each time-specific observation because the latter are only slightly dependent. In this article, we focus on its application.

We present two implementations of overarching Super Learners, called the discrete and continuous overarching Super Learners. Their predictions are generally accurate and better than those obtained (for comparison) by averaging all the predictions made by the base machine-learning algorithms. Specifically, the two Super Learners' predictions are quite good for all years except 2012 and 2016. The poorer predictions in 2016, a year known in the French insurance market to be particularly challenging, are more problematic because the real cost in 2016 is much higher than in 2012. Moreover, we were given the authorization to compare the predictions of the discrete and continuous overarching Super Learners with that of the algorithm currently deployed at CCR for the sole year 2017: the precisions are respectively 96% (discrete overarching Super Learner), 94% (continuous overarching Super Learners) and 83% (currently deployed algorithm).

In conclusion, the quality of the predictions made by the overarching Super Learners strongly depends on the quality of the local description of the drought event. It would probably benefit from a refined version of the city-level SWI that, contrary to the one we rely on, does not assume that the nature of the soil is the same all over France. In addition, the local description would also be considerably enhanced if it included information such as the distribution of proximity between a house and a tree at the city-level, or the distribution of the depth of house foundations at the city-level. Such pieces of information are proxies to the soil shrinkage and swelling. The local description could also be considerably enhanced by including direct measurements of soil shrinkage and swelling which can be obtained by radar interferometry.

In this work, we forecast the cost of drought events in France by super learning for those cities that have obtained the government declaration of natural disaster. The next step will be to predict which cities will obtain the government declaration of natural disaster. Tackling this difficult challenge will allow forecasting the cost of drought events earlier.

*Code availability.* The numerical analysis was conducted in R (R Core Team, 2022). We adapted the R package SuperLearner (Polley et al., 2021) in a package called SequentialSuperLearner (Chambaz and Ecoto, 2021).

*Author contributions.* The authors contributed equally to this work.

*Competing interests.* The authors contributed equally to this work.

*Acknowledgements.* The authors thank Thierry Cohignac (Caisse Centrale de Réassurance) for his suggestions.





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
