# Peer review of "Forecasting the cost of drought events in France by super learning"

_EGUsphere, 2022_

## Referee Comment (RC2)

In my opinion, this manuscript is not suitable for publication in its current form, particularly in a journal focused on natural-hazards research.

My primary concerns can be summarized as follows:

1. The manuscript is missing many key details, such that it is not possible for me to determine whether the methodological approach adopted is sound. For example:
   a. The data section does not describe the data used from different data sources and how these data compare in terms of resolution, quality etc. There is no attempt made to justify the variables included in the algorithm or their significance, leaving the reader uncertain as to whether exclusion or inclusion of more variables would have improved the performance of the algorithm. Arbitrary divisions of the data (e.g., proportions of buildings built are categorized in different time intervals) are not explained and supported.
   b. There is no attempt made to introduce the very basic high-level concepts of super learning to unfamiliar readers, even to explain the concept somewhat succinctly in the abstract. This is not appropriate, given that the targeted journal is focused on natural-hazard research. Furthermore, it seems to me (based on the description provided in Section 4.2) that the authors may be evaluating the performance of the algorithm based on training (rather than test) data, which would not be appropriate.
   c. There is no justification provided for the authors' exclusive focus on cities. Why not also include the costs of droughts in rural areas, when total drought costs are available (according to Figure 1)?
   d. How is inflation factored into the observed costs, particularly those from many years ago? How can future changes in exposure and population be integrated into future projections of drought costs from these algorithms? These questions should be answered clearly within the text.
2. If the problem being tackled is "less challenging" than that of a previous study (as implied by the authors in line 270), then I am doubtful on what (if anything) the present study is contributing to the state of the art in this field.
3. As seen in Figure 1, the claims data does not adequately represent the full cost of the droughts in any given year. If the purpose of the algorithm is to predict claims data, then this might be acceptable but if the purpose of the algorithm is to predict overall drought costs, then these do not seem reasonable training data to me.
4. I am generally concerned by the arbitrary equivalence of droughts and natural disasters. Droughts are not the only natural disasters that France suffers, yet this seems to be incorrectly implied in a number of cases:
   a. Inputs to the algorithm include indicators on whether there have been (successful) requests for government declarations of natural disasters –these declarations do not necessarily indicate the occurrence of a drought.
   b. Figure 5 shows errors for regions where natural disasters (rather than specifically droughts) occurred.

More minor (but still important) concerns:

1. I cannot find a precise description of the aim of the study in the Introduction. (This is implicit but should be explicit for clarity).
2. Figure 3: The real costs shown in this figure do not seem to align with those shown in Figure 1 (e.g., the 2017 cost of >900 million shown in Figure 3 is not found in Figure 1). So what real costs are being shown here?